# The Involvement of Polyunsaturated Fatty Acids in Apoptosis Mechanisms and Their Implications in Cancer

**DOI:** 10.3390/ijms241411691

**Published:** 2023-07-20

**Authors:** Mayra Montecillo-Aguado, Belen Tirado-Rodriguez, Sara Huerta-Yepez

**Affiliations:** 1Unidad de Investigacion en Enfermedades Oncologicas, Hospital Infantil de Mexico, Federico Gomez, Mexico City 06720, Mexico; mayramontecillo@gmail.com (M.M.-A.); abtirado81@gmail.com (B.T.-R.); 2Programa de Doctorado en Ciencias Biomédicas, Facultad de Medicina, Universidad Nacional Autónoma de Mexico (UNAM), Mexico City 04510, Mexico

**Keywords:** polyunsaturated fatty acids (PUFAs), apoptosis, cancer

## Abstract

Cancer is a significant global public health issue and, despite advancements in detection and treatment, the prognosis remains poor. Cancer is a complex disease characterized by various hallmarks, including dysregulation in apoptotic cell death pathways. Apoptosis is a programmed cell death process that efficiently eliminates damaged cells. Several studies have indicated the involvement of polyunsaturated fatty acids (PUFAs) in apoptosis, including omega-3 PUFAs such as alpha-linolenic acid, docosahexaenoic acid, and eicosapentaenoic acid. However, the role of omega-6 PUFAs, such as linoleic acid, gamma-linolenic acid, and arachidonic acid, in apoptosis is controversial, with some studies supporting their activation of apoptosis and others suggesting inhibition. These PUFAs are essential fatty acids, and Western populations today have a high consumption rate of omega-6 to omega-3 PUFAs. This review focuses on presenting the diverse molecular mechanisms evidence in both in vitro and in vivo models, to help clarify the controversial involvement of omega-3 and omega-6 PUFAs in apoptosis mechanisms in cancer.

## 1. Introduction

According to the World Health Organization (WHO), cancer is a significant global public health problem, with over 19,292,789 new cases and 9,958,133 deaths reported in 2020 [1]. Despite advancements in cancer therapy, conventional treatments often result in unsatisfactory survival rates. One of the primary processes affected in cancer is cell death pathways, and resistance to therapy often occurs due to the evasion of apoptosis [2]. Apoptosis is a programmed cell death process characterized by two main activation pathways: the extrinsic and intrinsic pathways [3]. There is substantial evidence suggesting that polyunsaturated fatty acids (PUFAs) play a crucial role in cancer development and can regulate various biological processes, including apoptosis [4,5,6]. Several studies have attributed a pro-apoptotic activity to omega-3 PUFAs, as they modulate the expression of key molecules involved in apoptosis induction [7,8]. For example, omega-3 PUFAs have been shown to upregulate miRNAs involved in the regulation of apoptotic genes in glioma cells [9]. On the other hand, the effect of omega-6 PUFAs remains controversial, with some authors indicating their role in inducing cell death; however, others suggest otherwise [10]. These findings are significant because PUFAs are considered essential fatty acids that are consumed through the diet [11]. Based on the first unsaturation and counting from the end of the alphatic chain, there are two types of PUFAs [12]: (1) omega-3 PUFAs, such as alpha-linolenic acid (ALA) (C18:3, n-3), eicosapentaenoic acid (EPA) (C20:5, n-3), and docosahexaenoic acid (DHA) (C22:6, n-3) and (2) omega-6 PUFAs, including linoleic acid (LA) (C18:2, n-6), gamma-linolenic acid (GLA) (C18:3, n-6), and arachidonic acid (ARA) (C20:4, n-6) [13].

Considering the high consumption of omega-6 PUFAs in the West, the modern Western diet (with an omega-6/omega-3 ratio of 20:1) [14] contains an excess of omega-6 PUFAs compared to Mediterranean diets (4:1) [15]; this is due to agribusiness and modern agriculture and have occurred in parallel with a significant increase in the prevalence of overweight, obesity and cancer [16]. The potential impact of the omega-6/omega-3 ratio on tumor behavior [17] and the regulation of cancer pathways, including apoptosis [10], makes it crucial to study the effects of these PUFAs on apoptosis pathways. 

In this review, we will examine the effects of omega-3 and omega-6 PUFAs on different apoptosis pathways reported in cell and animal models of various cancers. We will also analyze the molecular pathways involved in their pro-apoptotic and anti-apoptotic effects. This data supports the notion that consuming PUFAs, especially omega-3 PUFAs, can promote apoptotic pathways and may be applicable as an adjuvant therapy for cancer patients, in combination with other treatment modalities. However, the effect of omega-6 PUFAs, particularly ARA, is not yet fully understood.

## 2. Overview of Apoptosis Pathways

The apoptosis process is a physiological mechanism of programmed cell death. It involves two main pathways: (a) the extrinsic pathway and (b) intrinsic pathway. The extrinsic pathway is mediated by membrane cell-death receptors such as the Fas cell surface death receptor (FAS), TNF-related apoptosis-inducing ligand receptor (TRAIL-R), and TNF receptor superfamily member 1A (TNFR1). Upon binding to their ligands, these receptors recruit adapter proteins like the FAS-associated death domain protein (FADD) and TNFR-associated death domain protein (TRADD), forming the death-inducing signaling complex (DISC). The DISC facilitates the recruitment and activation of caspase-8 or -10 [18,19], (Figure 1A).

On the other hand, the intrinsic pathway is activated by various stimuli, including cytokine deprivation, reactive oxygen species (ROS) generation, DNA damage, and endoplasmic reticulum (ER) stress [20]. These triggers lead to mitochondrial outer membrane permeabilization (MOMP), which results in the release of mitochondrial proteins such as cytochrome c and Smac. The released proteins then activate initiator caspase-9 and executioner caspases-3, -6, and -7, leading to substrate cleavage, including Poly(ADP-ribose) polymerase (PARP), calcium-independent phospholipase A2 (iPLA2), and the destruction of subcellular structures [19,21]. The Bcl-2 family, comprising both pro-apoptotic (Bax and Bak) and anti-apoptotic (Bcl-2 and Bcl-xL) members, tightly regulates MOMP [22,23] (Figure 1B). Figure 1C shows that both pathways also exhibit crosstalk, with caspase-8 cleaving Bid to generate the active truncated form of Bid (tBid), which further triggers MOMP [24].

## 3. Omega-3 PUFAs and Apoptosis

Omega-3 PUFAs, including ALA, EPA, and DHA, play a crucial role in regulating various cellular functions such as membrane fluidity, protein functions, eicosanoid metabolism, gene expression, and cell signaling [25]. These fatty acids have been extensively studied for their potential in inhibiting the occurrence and progression of cancer. In fact, human studies have demonstrated that a high consumption of fish oil, rich in omega-3 PUFAs, can reduce the risk of cancer [26,27]. Furthermore, numerous studies have provided evidence supporting the role of omega-3 PUFAs in the regulation of apoptotic pathways, which is considered one of the biological mechanisms responsible for their positive effects. In this section, we will examine the involvement of omega-3 PUFAs in triggering apoptotic pathways.

### 3.1. Alpha-Linolenic Acid

ALA is an essential polyunsaturated fatty acid (PUFA) that cannot be synthesized by the human body and must be obtained from the diet [28]. It is found in leafy green vegetables, walnuts, flaxseeds, hemp, canola, and soybean oils [29]. Previous studies have demonstrated the anticarcinogenic potential of ALA. For instance, ALA has been shown to induce apoptosis through various molecular alterations. These include the accumulation of lipid droplets in the cytoplasm, lipid peroxidation, generation of ROS, induction of superoxide dismutase (SOD) activity in gastric carcinoma cells, (MGC and SGC) [30], and a mouse papilloma model [31]. 

Other studies have also suggested the participation of PUFAs as potent modulators of animal ion channels. They can form micelles that can fuse with the lipid bilayer, influencing cell membrane organization and altering channel function. PUFAs are known to increase membrane fluidity and screen surface charges, resulting in a shift in the voltage-dependence of the channels [32]. Particularly, ALA can cause accumulation and increase in intracellular calcium ion (Ca^2+^) levels, generating arrest in the sub-G1 phase and triggers the pro-apoptotic process [33,34]. Additionally, it has been observed that ALA provides a downregulation of the voltage-dependent anion channel (VDAC) present in the outer mitochondrial membrane, support the mitochondrial apoptosis [35]. 

These pieces of evidence have been demonstrated through changes in cell morphology, annexin V (AV) staining, disrupted mitochondrial function, caspase activation, and DNA fragmentation in various cell lines and mouse models [35,36,37,38]. Specifically, ALA activates the intrinsic pathway of apoptosis through mitochondrial depolarization, downregulation of anti-apoptotic proteins (Bcl-xL and Bcl-2), upregulation of pro-apoptotic proteins (Bad and Bax), Apaf, cytochrome c release, proteolytic activation of caspases-9 and -3, and proteolytic cleavage of PARP in ER+ MCF-7 breast cancer cells [33,35] and colon cancer cells (LoVo and RKO) [34]. Even a study involving different breast cancer cell lines suggests that ALA induces apoptosis regardless of estrogen receptor expression (ER) and estrogen (E2) environment [39] (Figure 2A).

Regarding the extrinsic pathway, one study reported that ALA induces an increased levels of caspase-8 in ER+ MCF-7 breast cancer cells [35]. Another study demonstrated an increase in Fas ligand (FasL), activation of caspases-8, -3, -7, and DNA damage as indicated by elevated p-H2A.X levels in human Jurkat cells [40] (Figure 2B). In summary, all this evidence supports the notion that the consumption of ALA primarily triggers the activation of mitochondrial apoptosis and to a lesser extent the extrinsic pathway, imparting anticancer effects. 

### 3.2. Eicosapentaenoic Acid

EPA can be obtained from ALA metabolism by a series of desaturation and elongation reactions [41]. Moreover, EPA is found in seafood like salmon, tuna, sardine, and marine oils [42]. EPA has important anti-cancer effects, mediated by the induction of expression of anti-inflammatory mediators, inhibition of cell proliferation, and modulation of cell death pathways [43]. The proposal mechanisms in the activation of apoptosis by EPA involve changes in numerous cell pathways. For example, induction of cell cycle arrest on breast cancer cell lines (MCF-7, SKBR-3, and MDA-MB-231) [44], particularly arresting the progression of the cell cycle from S to G2-M phase on BT20 breast tumor cells [45], downregulation of Akt/NFkB cell survival pathway on human breast cancer cell line MDA-MB-231 [46], in combination with DHA and vitamin D3 promote the upregulation of Raf/MAPK pathway [47], modification of lipid rafts and sustained activation of EGFR/p38MAPK pathway on MDA-MB-231 cells [48], in combination with estrogen increase GPER1/cAMP/PKA signaling on MCF-7 and T47D cells [49], inhibition of cholesterol biosynthesis inducer (SREBP2) and cholesterol efflux channel protein (ABCA1), causing cellular accumulation of cholesterol and subsequent increase in cell membrane polarity on TNBC cells [50]. Moreover, studies have shown that EPA induces inhibition of the phosphorylation of ERK1/2, Akt, and mammalian target of rapamycin (mTOR), blocking nuclear factor κB (NF-κB) p65 translocation from the cytoplasm into the nucleus on SKOV-3 cells [51]. This agrees with other authors who presented the inhibition of NFκB activity on multiple myeloma (MM) cells model [52,53]. Additionally, another main trigger is the high production of ROS after EPA treatment, which has been reported on breast cancer cells [44], bladder cancer cells [54], colorectal cancer cells [55], human glioma cells [56], and PC3 prostate cancer cells lines [57]. Zhang et al. described that ROS formation, leads to Ca^2+^ accumulation, the mitochondrial permeability transition pore opening, and JNK activation on the HepG2 cells [58]. Other less common mechanisms including Bcl-2 suppression through the p53/miR-34a axis on MM cells [59], wt-p53 accumulation on wt-p53 Molt-4 cells ALL cell line [60], inhibition of Akt/mTOR pathways through PPARγ/PTEN axis on MCF-7 cells [61], activation of acyl-CoA synthetase (ACS) on lymphoma cell line Ramos [62], and displacement of ARA from tumor-cell membranes after omega-3 PUFAs treatment on in vivo model [63]. 

Particularly, EPA activates the intrinsic pathway by loss of mitochondrial membrane potential leading to cytochrome c, and Smac/Diablo release to the cytosol, activation of caspase-3, 7, 9, cleavage PARP, and DNA fragmentation on several models including: human bladder cancer cells [54], lung cancer cells [64,65], carcinosarcoma cells [66], prostate cancer cell line [67], lymphoma cell line [68], esophageal squamous carcinoma cell lines [43], human pancreatic cancer cell lines [69,70], human colorectal cells [71], colorectal cancer stem cells [72], and even in the peripheral blood mononuclear cells (PBMCs)of MM patients in vitro [73]. Furthermore, it has been demonstrated, that EPA causes upregulation of pro-apoptotic proteins like Bax, Bak [74,75], and Adenosine A1 Receptor (ADORA1), which is a subtype of adenosine receptor functionally involved in cell death on gastric cancer cells [76]. In contrast, EPA promote downregulation of Bcl-xL [74], survivin [60], and X-linked inhibitor of apoptosis protein (XIAP) [75] (see Figure 3A). 

The available data indicates that apoptosis induced by EPA, also follows the extrinsic pathway, as shown in Figure 3B. Ewashuk et al. showed that EPA increased FAS surface expression on MDA-MB-231 cells [77]. Furthermore, Giros et al. reported downregulation of FLICE inhibitory protein (FLIP) on colorectal cancer cells [75]. Additionally, Fukui et al. also described that EPA caused caspase-8 dependent apoptosis on human pancreatic cancer cells MIA-PaCa-2 and Capan-2 [78]. This same effect was observed on human promyelocytic leukemia cells) [79], and breast tumor cells [45]. Moreover, evidence has shown that a crosstalk exists between both the extrinsic and intrinsic pathways mediated by EPA mainly through tBid [79]. Complementing this idea Giros et al. indicated that the timing of caspase-8 activation, and the oligomerization of Bid with Bax, support this crosstalk. It is worth noting that none of the mechanisms investigated were found to be responsible for triggering the apoptosis cascade induced by EPA [75]. Likewise, treatment with C20E (an analog of EPA) through TNFR1 activate ASK1-MKK4/7-JNK/p38MAPK pathway, promoted tBid, leading to MOMP and the activation of the mitochondrial pathway on human triple-negative MDA-MB-231 breast cancer cells and xenograft models [80] (see Figure 3C).

In summary, the research findings indicate that EPA can activate multiple molecular mechanisms, including both classical and alternative apoptotic pathways, and can regulate survival and cell growth signaling, ultimately leading to cell death in various cellular and animal models. However, further experiments are needed to explore the specific details of the extrinsic pathways, especially the upstream events. These findings suggest that EPA holds promise as a potential novel treatment for cancer, which could complement existing therapies.

### 3.3. Docosahexaenoic Acid

DHA is a marine-derived PUFA, that can be obtained from the metabolism of ALA [42,81]. Numerous studies have demonstrated the significant antitumor effects of DHA, including the suppression of neoplastic transformation, angiogenesis, and proliferation, as well as the induction of apoptosis in various cancer cells [65,82]. 

DHA exerts its proapoptotic effect by altering various molecular mechanisms. One of the key mechanisms involves modifications of the plasma membrane environment, which result in changes in the molecular composition, fluidity, structure, and function of lipid rafts [10]. These alterations impact the localization of cell surface receptors such as G protein-coupled receptors (GPCRs), Toll-like receptors (TLRs), retinoid X receptors (RXRs), and the epidermal growth factor receptor (EGFR), which play crucial roles in regulating cell growth signaling and apoptosis [83,84,85]. Another mechanism that affects the plasma membrane is the upregulation of syndecan-1 (SDC-1), a plasma membrane molecule that is induced by DHA through peroxisome proliferator-activated receptor-gamma (PPARγ) [86,87]. 

Additionally, the pro-apoptotic effect of DHA involves the modulation of several molecular survival pathways. For instance, DHA inhibits the Akt-mTOR axis by upregulating PPARγ and PTEN [61], induces AMPK activation [88], promotes mutant p53 expression and ROS generation [89], and downregulates the Akt/NFκB cell survival axis in MDA-MB-231 human breast cancer cells [46]. Moreover, DHA can induce apoptosis by upregulating MKP-1 phosphatase and downregulating p-ERK1/2 and p-p38 in lung cancer cells [90]. It can also downregulate the EGFR/STAT3/cyclin-D1/survivin and NF-κB/IκBα/cyclin-D1/survivin pathways, leading to apoptosis in pancreatic cancer cells (PANC-1) [91]. DHA activates the MAPKs (ERK/JNK/p38) axis [92,93], and in C57BL/6 Fat-1 transgenic mice with LLC cells, DHA induces apoptosis by activating AMPK and downregulating the PI3K/Akt signaling pathway [88]. Furthermore, DHA downregulates pSTAT3/Akt in renal cancer cells, resulting in apoptosis [94]. Notably, DHA induces apoptosis in human colorectal cancer cells (SW480 and HCT116) by inhibiting β-catenin protein expression through proteasomal degradation and reducing β-catenin translocation into the nucleus [95]. Also, DHA may induce apoptosis in cancer cells through the p53, MAPK, TNF, PI3K/Akt, and NFκB signaling pathways in human prostate cancer DU145 cells [96]. Another less explored mechanism suggests that DHA significantly increases the ratio of cyclic cAMP/cGMP levels and promotes Toll-like receptor 4 (TLR4) expression through PPARα, leading to breast cancer cell apoptosis [97]. DHA also induces cell cycle arrest by increasing the population of sub-G1 cells in glioblastoma cells [98], and human colon cancer cells [99]. 

Another important mechanism by which DHA triggers apoptosis is through oxidative stress resulting from the accumulation of ROS and lipid peroxidation [100]. DHA has been shown to stimulate ROS generation in various cell lines, including human promyelocytic leukemia cells (HL-60) [79], A549 cells [82], HT-29 cells [100], C6 glioma, SH-SY5Y neuroblastoma cell lines [93], cisplatin-resistant gastric cancer SNU-601/cis2 cells [101], docetaxel-resistant PC3 prostate cancer cells [102], and in an in vivo model of breast cancer [103]. Tsai et al. demonstrated that increased ROS production by DHA activates the PI3K/Akt/Nrf2 signaling pathway and induces the expression of Oxidative stress-induced growth inhibitor 1 (OSGIN1) [104]. Similar, Nrf2 nuclear translocation in response to oxidative stress has been observed in AML cell lines (U937, MOLM-13, and HL-60) [105]. Furthermore, downregulation of antioxidant enzymes such as catalase (CAT) in A549 cells [82] and glutathione (GSH) in the human PaCa-44 pancreatic cancer cell line has been reported to contribute to the accumulation of intracellular ROS [106]. However, Geng et al. demonstrated that DHA treatment increased the activity levels of major antioxidant enzymes, including total superoxide dismutase (t-SOD), CAT, and glutathione peroxidase (GSH-PX), while decreasing the concentration of malondialdehyde (MDA) in human malignant breast tissues [97]. The effect of DHA on antioxidant enzymes is not completely clear, and discrepancies observed may be related to the cell models, requiring further research. Another proposed mechanism is that DHA causes inactivation of prostaglandin family genes, lipoxygenases, and alters the expression of PPARα and γ, leading to lipid peroxidation in CaCo-2 colon cancer cell lines [107].

Furthermore, recent innovative studies have focused on how DHA exerts a pro-apoptotic effect through the activation of ER stress. It is well-known that conditions interfering with ER function lead to the accumulation and aggregation of unfolded proteins, which can trigger apoptotic cell death if the stress is prolonged or the adaptive response fails [108]. Through RNA-seq analysis of MDA-MB-231 cells treated with DHA, Chénais et al. showed that DHA downregulates genes involved in the cholesterol biosynthesis pathway and upregulates genes associated with ER stress response, including NEF, BiP, HSP40, GADD34, ATF4, IRE1a, XBP1, CHOP, ERO1B, SEL1L, HERPUD1, and HSPA6 [109]. Similarly, Jakobsen et al. demonstrated that DHA induces the expression of various factors involved in ER stress response, such as XBP1, PERK, ATF4, ATF6, and phosphorylated EIF2a, in SW620 colon carcinoma cells [110]. This discovery has also been observed in cisplatin-resistant gastric cancer SNU-601/cis2 cells [101]. 

The events ultimately lead to the activation of the intrinsic pathway, which results in the depolarization of the mitochondrial membrane. This process involves the downregulation of Bcl-xL and Bcl-2, the upregulation of Bax and Bcl-xS, and the dimerization of Bax and Bak [75,111]. These events lead to MOMP and the release of Smac/Diablo and cytochrome c from the mitochondria. Subsequently, caspase-9 and caspase-3/7 are activated, leading to PARP cleavage and the formation of DNA adducts in various cell lines [93,98,99,104,112,113,114,115,116,117], and in vivo models [118,119]. Additionally, DHA downregulates XIAP [75] and survivin [72,120,121]. DHA also increases the accumulation of wt-p53 and the expression levels of microRNA-16-1 (miRNA-16-1) in chemotherapy-resistant colorectal cancer stem cells with a KRAS mutation [121]. Other apoptotic effects observed with DHA treatment include DNA fragmentation, phosphatidylserine externalization, and mitochondrial membrane depolarization in human neuroblastoma LA-N-1 cells [74]. In line with these findings, Sun et al. demonstrated that DHA upregulates the gene expressions of Caspase-9, Caspase-3, DFFA, TP53, BAX, CIDEA, TNF, TNFRSF1A, Caspase-1, and LTA, while downregulating XIAP, AIFM1, BIRC6, AKT1, and BID in human prostate cancer DU145 cells [96] (see Figure 4A).

Other studies also support the activation of the extrinsic pathway by DHA (Figure 4B), which involves an increase in death receptors such as TRAIL, death receptor 4 (DR4), and FAS in MCF 7 breast cancer cells [115]. Additionally, Ewaschuk et al. demonstrated that DHA increases FAS surface expression and induces the movement and raft clustering of FAS and FADD in the cell membrane of MDA-MB-231 cells [77]. In contrast, Giros et al. did not find any of the death receptors (FAS, TNFR1, and TRAIL-R2) to be responsible for triggering the apoptosis cascade induced by DHA. However, they proposed that the downregulation of FLIP is responsible for cell death [75]. Further research is required to elucidate the effect of DHA on death receptors. Moreover, the effect of DHA on different death ligands has also been investigated. Particularly, DHA sensitizes the apoptotic response to TNF-α and anti-FAS antibody (CH-11) in human colon adenocarcinoma HT-29 cells [100]. It has also been reported that DHA combined with TRAIL treatment triggers the extrinsic pathway, causes ER stress, decreases XIAP and cIAP1 levels, and alters sphingolipid metabolism, inducing apoptosis in human colon cancer cells [122,123]. Similarly, Fluckiger et al. proposed a novel indirect mechanism in which treatment with DHA leads to the nuclear accumulation of Foxo3a, which binds to the microRNA-21 (miR-21) promoter, triggering its transcriptional repression. This, in turn, increases TNFα mRNA levels and induces apoptosis in an autocrine manner in human colorectal cancer cells [124]. These events can activate caspase-8 and executioner caspases, thereby promoting apoptosis [79,103]. Similar to EPA, DHA can also initiate a crosstalk between the extrinsic and intrinsic apoptotic pathways through Bid cleavage (Figure 4C) [75,79,122,123]. 

In conclusion, it is important to highlight that DHA induces apoptosis through direct or indirect pathways, involving the suppression of survival pathways and the activation of extrinsic and intrinsic death pathways. These findings demonstrate that DHA has the ability to induce cell apoptosis, offering promising and safe options for cancer treatment in the future.

## 4. Omega-6 PUFAs and Apoptosis

In Western populations, there has been an observed increase in the consumption of omega-6 PUFAs, resulting in a skewed omega-6/omega-3 PUFA ratio of 20:1 [125]. This imbalance has been suggested to contribute to the rising incidence of cardiovascular, inflammatory, and oncological diseases [14]. Mounting evidence indicates that the adverse effects associated with omega-6 PUFAs may be attributed to arachidonic acid (ARA). On the other hand, linoleic acid (LA) and γ-linolenic acid (GLA) have demonstrated certain anti-cancer activities, including the induction of apoptosis [126]. In this section, we will present and discuss recent findings on the potential effects of omega-6 PUFAs in apoptotic processes.

### 4.1. Linoleic Acid

Like ALA, LA is an essential polyunsaturated fatty acid that humans can only obtain from their diet [28]. LA is the most abundant PUFA in nature, comprising 50% to 80% of the fatty acids found in vegetable oils such as soybean, sunflower, safflower, and corn oils [127,128]. In the Western diet, LA is the primary PUFA, accounting for over 85% of PUFA intake [129]. Numerous pieces of evidence suggest that LA may be involved in various pathological processes, including cancer. Specifically focusing on cell death processes, the data indicates that LA has a proapoptotic effect. 

The proposed mechanisms for LA-induced apoptosis involve the accumulation of lipid droplets and increased lipid peroxidation in gastric carcinoma cells [30], and colon cancer cells [130]. Additionally, an LA isomer called alpha-eleostearic acid (α-ESA), which possesses a conjugated triene system, has been associated with the inhibition of survival pathways. For example, it downregulates ERK1/2 through PPARγ in MCF-7 breast cancer cells [131], and inhibits the Akt/GSK-3β survival pathway by activating PTEN in SKBR3 and T47D breast cancer cell lines [132]. Another LA isomer, beta-eleostearic acid (β-ESA, 9E11E13E-18:3), induces apoptosis through oxidative stress, leading to the accumulation of ROS and a decrease in GSH in T24 human bladder cancer cells [128], as well as in gastric carcinoma cells [30]. Moreover, LA induces ROS production on HepG2 Cells [133]. The involvement of ER stress has also been investigated, with studies reporting intracellular calcium (CA^2+^) accumulation and the expression of unfolded protein response (UPR)-associated genes (CHOP, GRP78, and GRP94) in hepatoma H4IIE cell lines [134]. Another reported mechanism is the downregulation of prostaglandin E2 (PGE2) production and telomerase activity through the suppression of cyclooxygenase 2 (COX-2) expression in gastric adenocarcinoma AGS cells [135]. 

Through the aforementioned mechanisms, LA triggers the mitochondrial pathway by causing a loss of mitochondrial membrane potential, upregulating the expression of Bax and Bad, and downregulating the expression of Bcl-2. This leads to MOMP and the release of cytochrome c, subsequently activating caspase-9 and caspase-3 and reducing ATP levels in various cancer cell lines (Figure 5A) [130,131,132,136,137,138,139]. Regarding the extrinsic pathway, Muzio et al. reported that conjugated linoleic acid (CLA), a term used for a series of isomers of linoleic acid, increases caspase-8 levels in human hepatoma SK-HEP-1 cells [140]. Liu et al. also demonstrated enhanced expression of FAS, a death receptor, in the SGC-7901 gastric adenocarcinoma cell line when exposed to the c9,t11 isomer of CLA [141], suggesting the induction of the extrinsic apoptotic pathway, as shown in Figure 5B. However, further research is needed to elucidate the detailed mechanisms of activation.

All this evidence indicates that LA plays a crucial role in inducing apoptosis. Several lines of evidence suggest that the effect is primarily mediated through the intrinsic pathway, while more research is needed on this subject. 

### 4.2. Gamma-Linolenic Acid

GLA is synthesized in the body from LA through the action of the enzyme D-6-desaturase. While small amounts of GLA can be obtained from green leafy vegetables and nuts, it is primarily produced endogenously [10]. Numerous pieces of evidence suggest that GLA exhibits anti-cancer activities both in laboratory studies and animal models [126]. Additionally, several studies have demonstrated the influence of GLA and its metabolites on the expression of various genes and proteins involved in the apoptotic process [10]. 

The primary mechanism proposed for the induction of apoptosis by GLA involves oxidative stress resulting from the accumulation of ROS and lipid peroxidation. Colquhoun et al. showed that GLA increases ROS and lipid peroxide production, while decreasing the activity of mitochondrial respiratory chain complexes I+III and IV and mitochondrial membrane potential in carcinosarcoma cells [142]. Similarly, high levels of MDA, a marker of lipid peroxidation, have been observed in leukemia cells treated with GLA, and the cytotoxic effects of GLA were blocked by the antioxidant BHT [143,144]. GLA has also been found to inhibit mitochondrial carnitine palmitoyltransferase I (CPT I) in Hep2 human larynx tumor cells, leading to fatty acid oxidation [145]. Another pathway implicated in GLA-induced apoptosis is the ROS/ASK1/JNK/p38 MAPK axis in KF28 ovarian cancer cells [146]. Furthermore, GLA has been shown to induce cell cycle arrest and sub-G1 accumulation in ER+ MCF-7 cells and lymphoblast cell lines (TK6 and WTK1) [35,147].

All of the aforementioned mechanisms contribute to intrinsic apoptosis by downregulating the expression of Bcl-xL and Bcl-2, upregulating Bad, leading to the loss of mitochondrial membrane potential, release of cytochrome c, activation of caspase-9 and -3, cleavage of PARP, DNA fragmentation in various cell lines [142,148,149,150], as well as in in vivo models [56,66,149,151]. The involvement of caspases in GLA-induced apoptosis has been confirmed by the inhibition of apoptosis through the use of a pan-caspase inhibitor (z-VAD-fmk) in leukemic cells [144]. Morphological and physiological changes indicative of apoptosis, such as nuclear staining with DNA-binding fluorochrome Hoechst 33342, chromatin condensation, nuclear fragmentation [143], and TUNEL-positive cells [56,148,150], have further supported the apoptotic effect of GLA (Figure 6).

Taken together, these findings suggest that GLA predominantly induces intrinsic apoptotic pathway through mechanisms associated with the generation of oxidative stress. However, information regarding the involvement of GLA in the extrinsic pathway is limited, and further investigations are required to determine its role in the pathways that trigger extrinsic apoptosis.

### 4.3. Arachidonic Acid

ARA is typically esterified to membrane phospholipids and is one of the most abundant PUFA present in human body [152]. ARA consumption is very abundant in our daily diet, particularly in the Western diet, and various foods contain high concentrations of ARA, like eggs, lean meat and meat fats of beef, lamb, pork, chicken, duck, and turkey [153,154]. Additionally, ARA is synthesized by a series of desaturases and elongases from LA [155]. Plenty of studies have associated the high ARA intake with many adverse effects on the human body, including cancer promotion, mainly attributed to its metabolites like PGE2 [126]. However, increasing evidence also suggests that ARA has ability to enhance the cytotoxic action of various anti-cancer drugs and possess certain antitumoral activities, including proapoptotic effect [10]. In this context, it is noteworthy that ARA presents a contradictory role in cancer progression. 

To understand this controversial effect, it is crucial to examine the molecular apoptotic pathways regulated by ARA. As shown in Figure 7A, the mechanisms proposed in the activation of apoptosis by ARA included the increased oxidative stress generated by ROS production and lipid peroxidation, high levels of MDA, 4-hydroxy-2-nonenal (4-HNE), high activity of SOD, and GSH-PX, as well as downregulation of GSH on several cancer cell lines [156,157,158,159,160]. In the same way, elevation of ROS levels and intracellular Ca^2+^ concentration trigger activation of p38α MAPK and JNK1 pathways on human neuroblastoma SK-N-SH cells [161]. On the other hand, Bae et al. showed that ARA increases the processed form of XBP1 (Pxbp1) and phosphorylated Eif2α (p-Eif2α) triggering ER stress on HT-29 human colon cancer cells [162]. Moreover, another mechanism reported, involves the high accumulation of unmetabolized ARA which activates the enzyme sphingomyelinase (Smase), this produces elevated ceramide levels that induces apoptosis [163]. For this reason, several studies have used inhibitors of enzymes involved in ARA metabolism such as Cpla2, COX-2, CYP4A, fatty acid coenzyme-A ligase 4 (FACL-4), coenzyme-A independent transacylase (CoA-IT), 5-LOX, leading to accumulation of intracellular ARA, elevation ceramide levels, and induction of apoptotic cell death on several cancer cell lines [164,165,166,167]. ARA mediates the mitochondria-dependent death pathway through previous stimuli, causing downregulation of Bcl-Xl and Bcl-2, as well as the loss of potential in the mitochondrial membrane, release of cytochrome c, accompanied with activation of caspase-9 and -3 that trigger the breakdown of PPAR. As well of externalization of phosphatidylserine and condensation of chromatin, on AS-30D rat hepatoma cells [168], Y79 retinoblastoma cells [160], HT-29 human colon cancer cells [162], and LoVo and RKO cells [130], among others. Furthermore, studies have reported typical morphological changes of programmed cell death such as pyknosis, karyorrhexis, cell-shrinkage and -blebbing were observed [158]. The proapoptotic effect of ARA has been supported by other experiments such as mitochondrial permeability assay, TUNEL labeling, and DNA laddering experiments [149,157]. 

On the other hand, the evidence of ARA’s participation in the extrinsic pathway is limited. In this regard, Polavarapu et al. demonstrated that the administration of ARA plus bleomycin increased the expression of FAS, caspases-8 and -3 in IMR-32 human neuroblastoma cells, suggesting the activation of the extrinsic apoptotic pathway [169]. However, more research is needed to fully elucidate this pathway (Figure 7B).

In contrast to the aforementioned evidence, many reports propose an inhibitory role of ARA in apoptosis. It is worth mentioning that ARA can also activate survival aproliferation pathways [13] which may prevent the activation of apoptosis. For example, Yang et al. showed that ARA can activate the Akt survival pathway in ovarian cancer cells [170]. Other authors have mentioned that metabolites derived from ARA, such as PGE2, PGE4, 12-HETE, and 20-HETE, are responsible for inhibiting apoptosis. Liu et al. reported that 12-HETE inhibits cell apoptosis in ovarian carcinoma cells through the activation of the integrin-linked kinase (ILK)/NF-κB axis [171]. Additionally, Cui et al. proposed a novel mechanism in which active caspase-3 can activate cytosolic calcium-independent phospholipase A2 (ciPLA2), leading to the release of ARA and the production of PGE2. This abnormal activation of focal adhesion kinase (FAK) can eventually accelerate the proliferation of SKOV3 ovarian cancer cells, suggesting that excessive apoptosis can have a negative effect [172] (see Figure 7C).

Interestingly, our recent research has demonstrated that a high intake of omega-6 PUFAs in a lung cancer model leads to a decrease in the expression of active caspases-3, -8, and -9. This decrease in caspase expression is associated with a corresponding increase in cell proliferation, as indicated by both the mitotic index and the expression of mini-chromosome maintenance protein 2 (MCM2) [17]. What makes these findings particularly intriguing is that despite the high concentration of omega-6 PUFAs in the diet (with an omega-6/omega-3 ratio of 20:1), our results are consistent with studies that support the idea of ARA, an omega-6 PUFAs, having an anti-apoptotic role. It is important to note that omega-6 PUFAs are essential fatty acids that play critical roles in various biological processes. However, maintaining a balanced ratio between omega-6 and omega-3 fatty acids is essential for overall health. The optimal ratio between these two types of fatty acids is still a subject of ongoing research and debate, but most experts suggest a ratio closer to 4:1 or even lower for maximum health benefits [173]. These findings highlight the complex relationship between omega-6 PUFAs, apoptosis, and cell proliferation specifically in the context of lung cancer. Further research is needed to better understand the underlying mechanisms involved and to evaluate the implications for human health.

Upon analyzing all the studies on ARA, the evidence presented shows that ARA can elicit both pro- and anti-apoptotic effects. The observed differences may be related to the specific models or concentrations of this PUFA used in the studies. The conflicting results mentioned above suggest that further research is required to fully understand the correlation between ARA and the apoptotic process.

## 5. Concluding Remarks and Future Perspectives

In conclusion, this review has provided a comprehensive compilation of information regarding the modulation of apoptotic pathways by PUFAs, which are essential components of a daily diet. Overall, PUFAs have been shown to have a proapoptotic effect, except for ARA, which has a dual role with both anti-apoptotic and pro-apoptotic effects. The evidence supports the notion that PUFAs play a significant role in the induction and sensitization of apoptosis in various tumor cells and murine cancer models. Considering their wide availability and lack of toxic effects, PUFAs may represent a promising and novel strategy for cancer therapy.

In addition, based on the comprehensive information presented in this review, we have a strong belief that omega-3 polyunsaturated fatty acids (PUFAs) exert a clear pro-apoptotic effect. On the other hand, omega-6 PUFAs, particularly arachidonic acid (ARA), exhibit a dual effect, including both pro-apoptotic and pro-survival outcomes. However, these behaviors appear to vary depending on the specific in vitro and in vivo models studied, the concentration of fatty acids used, the duration of incubation, and the combination of treatments with other drugs.

These findings highlight the complexity of the effects of PUFAs on apoptosis, suggesting that the specific outcomes are influenced by various factors. The context-dependent nature of these effects underscores the need for further research to elucidate the precise mechanisms underlying the diverse actions of PUFAs in apoptosis regulation. By considering these factors, future studies can provide valuable insights into optimizing the therapeutic potential of PUFAs in cancer treatment.

We also emphasize the importance of maintaining a balance between omega-6 and omega-3 fatty acid consumption, as it can lead to favorable biological outcomes, including decreased inflammation and the induction of apoptosis, which have a beneficial effect in preventing cancer development. Furthermore, the use of omega-3 polyunsaturated fatty acids (PUFAs) as adjuvant therapy in cancer treatment has been suggested by various researchers. Omega-3 supplementation in the diets of cancer patients undergoing chemotherapy has shown potential benefits, such as improved treatment tolerance, decreased tumor size, and increased lean body mass [174]. For example, in a randomized trial with 11 colorectal cancer patients, the group that ingested 2 g/day of fish oil exhibited significant increases in EPA and DHA levels in blood plasma (1.8 and 1.4 times higher, respectively), along with muscle mass gain (mean of +1.2 kg). This group also demonstrated greater chemotherapy tolerance and a reduction in tumor size compared to the untreated group [175]. In another study, 128 patients with gastrointestinal cancer and cachexia were provided with a diet containing or lacking 1.1 g of EPA, 0.5 g of DHA, and 16 g of protein. The patients were monitored using bioelectrical impedance analysis, and the authors observed that the fish oil-enriched diet contributed to improved chemotherapy tolerance, tumor shrinkage, and increased lean body mass over time [176].

Importantly, the use of enzyme inhibitors to enhance the accumulation of specific oxylipins, exerting pro-apoptotic effects, has shown promise in in vitro and murine models [177,178]. These findings indicate the potential of such inhibitors as powerful tools in cancer treatment. However, further research and clinical trials are necessary to validate their efficacy and safety in human patients.

Looking ahead, future perspectives in this field should aim to further elucidate the molecular mechanisms underlying the apoptotic effects of PUFAs, including the specific pathways involved and the interplay between different PUFAs. Additionally, more studies are needed to investigate the effects of PUFAs on different types of cancer cells and animal models, as well as to evaluate their potential synergistic effects with existing cancer therapies. Furthermore, clinical trials are warranted to assess the efficacy and safety of PUFAs as adjuvant or standalone therapies for cancer treatment. Overall, continued research in this area has the potential to uncover valuable therapeutic strategies and contribute to the development of novel approaches in cancer therapy. 

## Figures and Tables

**Figure 1 ijms-24-11691-f001:**
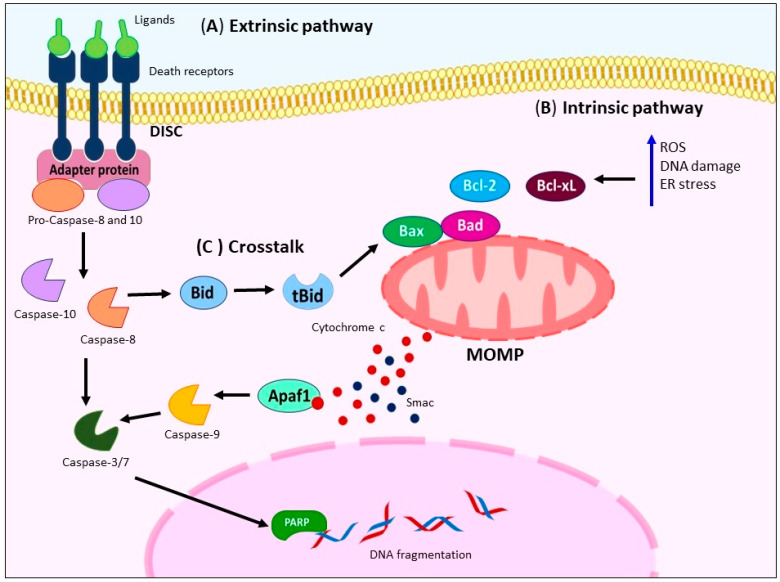
Principal mechanism of apoptosis activation. (**A**) Activation of extrinsic pathway occurs through ligand binding to death receptors (FAS, TRAIL-R, or TNFR1), subsequently mediates bind of adapter proteins (FADD or TRADD) building DISC, activating caspase-8/-10 and -3. (**B**) Several stimuli trigger intrinsic pathway like cytokine deprivation, increase generation of ROS, DNA damage, and ER stress (blue arrow), causing MOMP that leads to the release of the mitochondrial proteins [cytochrome c (red dots), Smac (blue dots)], and subsequent activation of initiator caspase-9 and executioners caspases-3, -6, and -7 concluding in apoptosis. (**C**) There is crosstalk between the extrinsic and intrinsic pathways through the cleavage of caspase-8, which generates a tBid. tBid triggers MOMP, amplifying the apoptotic signal. FAS: Fas cell surface death receptor. TRAIL-R: TNF-related apoptosis-inducing ligand receptor. TNFR1: TNF receptor superfamily member 1A. FADD: FAS-associated death domain protein. TRADD: TNFR-associated death domain protein. DISC: Death-inducing signaling complex. ROS: Reactive oxygen species. DNA: Deoxyribonucleic acid. ER: Endoplasmic reticulum. MOMP: Mitochondrial outer membrane permeabilization. tBid: Truncated form of Bid.

**Figure 2 ijms-24-11691-f002:**
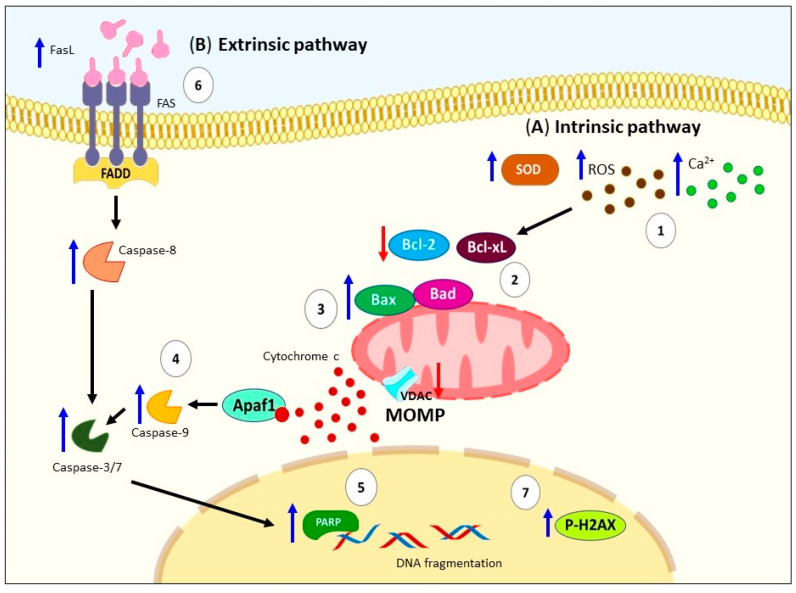
Schematic representation of apoptosis activation by ALA. (**A**) Several signals can trigger the activation of intrinsic pathway by ALA, for example: (**1**) generation of ROS, induction of SOD activity, and increase in intracellular Ca^2+^; (**2**) these signals promote mitochondrial depolarization, downregulation of antiapoptotic proteins (Bcl-xL and Bcl-2); (**3**) upregulation of proapoptotic proteins (Bad and Bax), VDAC, Apaf, as well as cytochrome c release; (**4**) proteolytic activation of caspases -9 and -3; and (**5**) proteolytic cleavage of PARP. (**B**) Regarding the extrinsic pathway, there has been reported (**6**) high levels of FasL, activation of caspases-8, -3, -7, and DNA damage caused by (**7**) p-H2A.X. ALA: Alpha-linolenic acid. ROS: Reactive oxygen species. SOD: Superoxide dismutase. Bcl-xL: B-cell lymphoma-extra-large. Bcl-2: B-cell lymphoma 2. Bad: Bcl-2 associated agonist of cell death. Bax: Bcl-2 Associated X-protein. VDAC: Voltage-dependent anion channel. PARP: Poly(ADP-ribose) polymerase. FasL: Fas ligand. DNA: Deoxyribonucleic acid. P-H2A.X: Phospho-Histone H2A.X. MOMP: Mitochondrial outer membrane permeabilization.

**Figure 3 ijms-24-11691-f003:**
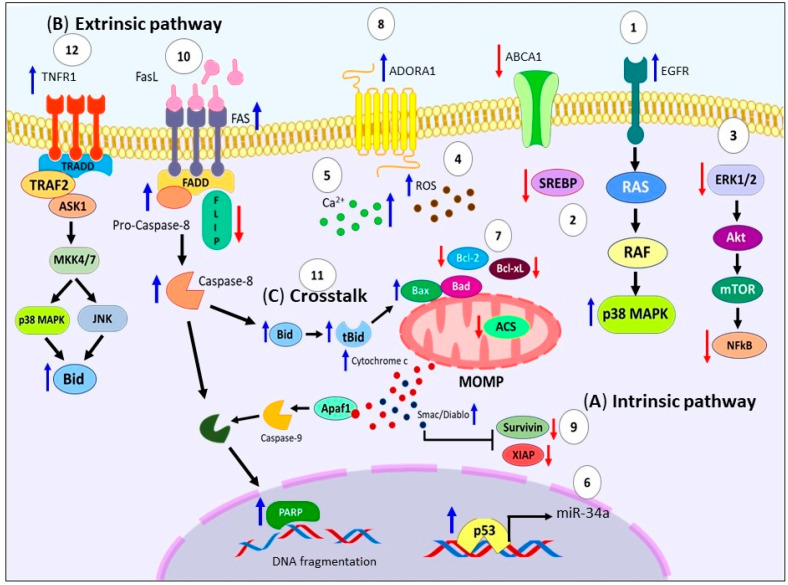
Molecular pathways of EPA-induced apoptosis in cancer models. (**A**) EPA can trigger intrinsic pathway activation through arresting the progression of the cell cycle, (**1**) upregulation of Raf/MAPK pathway, sustained activation of EGFR/p38 MAPK axis, (**2**) accumulation of cholesterol by inhibition of SREBP2 and ABCA1, (**3**) downregulation of survival pathways (ERK1/2/Akt/mTOR/NFkB), (**4**) generation of ROS, (**5**) intracellular Ca^2+^ accumulation, and (**6**) Bcl-2 suppression through the p53/miR-34a axis, wt-p53 accumulation, activation of ACS. (**7**) These signals promote mitochondrial depolarization, and cytochrome c and Smac/Diablo release to the cytosol, activation of caspases -9, -3, -7, cleavage PARP, and DNA fragmentation. (**8**) Also, EPA causes upregulation of Bax, Bak and ADORA1. (**9**) In contrast, EPA promote downregulation of Bcl-xL, Survivin and XIAP. (**B**) Regarding the extrinsic pathway, (**10**) EPA increased FAS surface expression, downregulate FLIP levels, and promote activation of caspase-8. (**C**) (**11**) Numerous studies demonstrate crosstalk between both pathways through caspase-8 cleaved that generates tBid, which can be oligomerized with Bax and that triggers MOMP. Moreover, (**12**) treatment with C20E through TNFR1 activate ASK1-MKK4/7-JNK/p38MAPK pathway, promoting the tBid. EPA: Eicosapentanoic acid. Raf: Rapidly accelerated fibrosarcoma. MAPK: Mitogen-activated protein kinase. EGFR: Epidermal growth factor receptor. SREBP2: Cholesterol biosynthesis inducer. ABCA1: Cholesterol efflux channel protein. ERK1/2: Extracellular signal-regulated protein kinases 1 and 2. Akt: Protein kinase B. mTOR: Mammalian target of rapamycin. NFkB: Nuclear factor κB. ROS: Reactive oxygen species. Bcl-2: B-cell lymphoma 2. p53: Tumor protein P53. miR-34a: MicroRNA 34a. wt-p53: Wild type Tumor protein P53. ACS: Acyl-CoA synthetase. PARP: Poly(ADP-ribose) polymerase. DNA: Deoxyribonucleic acid. Bax: Bcl-2 Associated X-protein. Bak: Bcl-2 homologous antagonist/killer. ADORA1: Adenosine A1 Receptor. Bcl-xL: B-cell lymphoma-extra-large. XIAP: X-linked inhibitor of apoptosis protein. FAS: Fas cell surface death receptor. FLIP: FLICE inhibitory protein. tBid: Truncated form of Bid. MOMP: Mitochondrial outer membrane permeabilization. C20E: 17,18-epoxyeicosanoic acid. TNFR1: TNF receptor superfamily member 1A. ASK1: Apoptosis signal-regulating kinase 1. MKK4/7: MAP2K4 mitogen-activated protein kinase kinase 4. JNK: c-Jun N-terminal kinase.

**Figure 4 ijms-24-11691-f004:**
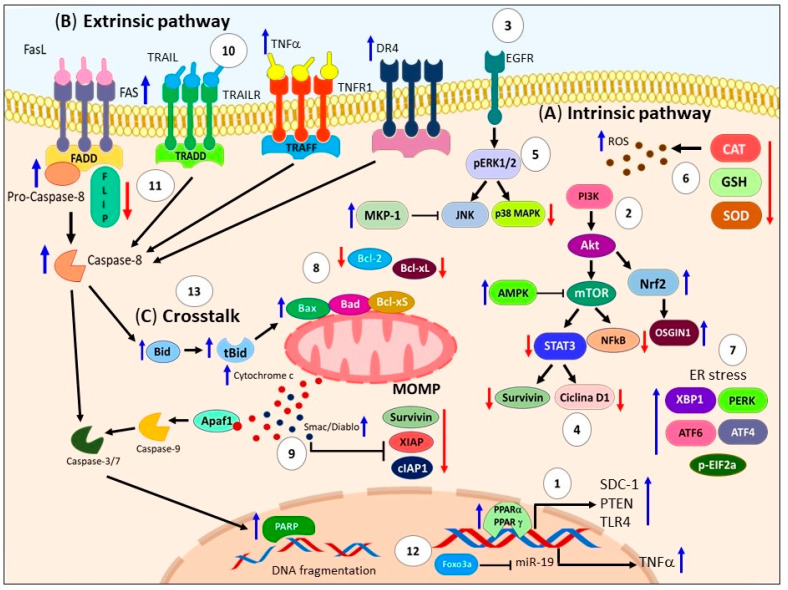
Illustrates the apoptosis mechanisms triggered by DHA. (**A**) DHA induces modifications in the plasma membrane environment through various pathways. (**1**) It increases the expression of SDC-1 via PPARγ, (**2**) inhibits the Akt-mTOR axis by upregulating PPARγ and PTEN, downregulates the Akt/NFκB cell survival axis, (**3**) EGFR/STAT3/cyclin D1/survivin and (**4**) NF-κB/IκBα/cyclin D1/survivin pathways, (**5**) activates the ERK/JNK/p38 axis, stimulates AMPK, and downregulates the PI3K/Akt pathway. (**6**) EPA also leads to increased ROS production, activates the PI3K/Akt/Nrf2 signaling pathway, and induces the expression of OSGIN1. The downregulation of antioxidant enzymes such as CAT, GSH, and SOD contributes to the accumulation of intracellular ROS. Furthermore, (**7**) EPA activates ER stress through the expression of XBP1, PERK, ATF4, ATF6, and phosphorylated EIF2a. These events collectively induce intrinsic apoptosis by triggering the depolarization of the mitochondrial membrane. (**8**) This is achieved through the downregulation of Bcl-xL and Bcl-2, upregulation of Bax and Bcl-xS, and dimerization of Bax and Bak (**9**) Consequently, MOMP occurs, leading to the release of Smac/Diablo and cytochrome c from the mitochondria. Activated caspase-9 and -3/7 further execute apoptosis by cleaving PARP and forming DNA adducts. DHA downregulates XIAP, cIAP1, and survivin. (**B**) On the other hand, (**10**) DHA triggers the extrinsic pathway of apoptosis by increasing the expression of death receptors such as TRAIL, TNFR, DR4, and FAS. Subsequently, (**11**) DHA activates FADD and downregulates FLIP. Moreover, DHA sensitizes the apoptotic response to death ligands such as TNF-α, FAS antibody, and TRAIL. Additionally, (**12**) DHA induces nuclear accumulation of Foxo3a, which binds to the miR-21 promoter, leading to its transcriptional repression and increased TNFα mRNA levels. These events ultimately activate caspase 8 and executioner caspases, promoting apoptosis. (**C**) (**13**) Both EPA and DHA demonstrate crosstalk between the intrinsic and extrinsic apoptotic pathways through tBid. These findings highlight the multifaceted apoptotic mechanisms induced by DHA, involving both direct and indirect pathways, and emphasize its potential as a safe and effective option for cancer treatment. DHA: Docosahexaenoic acid. SDC-1: Syndecan-1. PPARγ: Peroxisome proliferator-activated receptor-gamma. Akt: Protein kinase B. mTOR: Mammalian target of rapamycin. PTEN: Phosphatase and tensin homolog. NFκB: Nuclear factor κB. EGFR: Epidermal growth factor receptor. STAT3: Signal transducer and activator of transcription 3. IκBα: Nuclear factor of kappa light polypeptide gene enhancer in B-cells inhibitor, alpha. ERK: Extracellular signal-regulated kinase. JNK: c-Jun N-terminal kinase. AMPK: AMP-activated protein kinase. PI3K: Phosphoinositide 3-kinase. ROS: Reactive oxygen species. Nrf2: Nuclear factor erythroid 2-related factor 2. OSGIN1: Oxidative stress-induced growth inhibitor 1. CAT: Catalase. GSH: Glutathione. SOD: Superoxide dismutase. ER: Endoplasmic reticulum. XBP1: X-box-binding protein 1. PERK: Protein kinase R-like endoplasmic reticulum kinase. ATF4: Activating Transcription Factor 4. ATF6: Activating Transcription Factor 6. EIF2a: Eukaryotic translation initiation factor 2A. Bcl-xL: B-cell lymphoma-extra-large. Bcl-2: B-cell lymphoma 2. Bax: Bcl-2 Associated X-protein. Bcl-xS: Bcl-2 homologous short isoform. Bak: Bcl-2 homologous antagonist/killer. MOMP: Mitochondrial outer membrane permeabilization. DNA: Deoxyribonucleic acid. XIAP: X-linked inhibitor of apoptosis protein. cIAP1: Cellular inhibitor of apoptosis protein 1. TRAIL: TNF-related apoptosis inducing ligand. TNFR: Epidermal growth factor receptor. DR4: Death receptor 4. FAS: Fas cell surface death receptor. FADD: FAS-associated death domain protein. FLIP: FLICE inhibitory protein. TNF-α: Tumor necrosis factor-alpha. Foxo3a: Forkhead transcription factor O subfamily member 3a. miR-21: MicroRNA 21. mRNA: Messenger ribonucleic acid. tBid: Truncated form of Bid.

**Figure 5 ijms-24-11691-f005:**
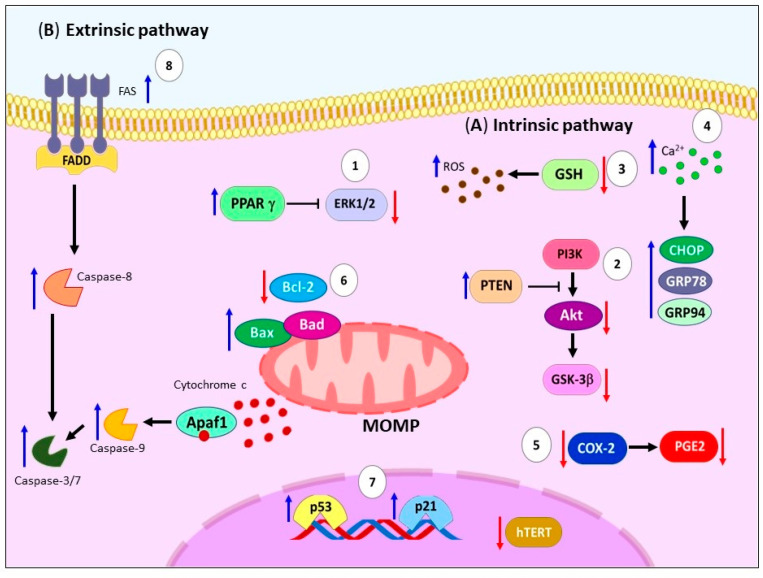
Molecular pathways of apoptosis regulate by LA. (**A**) LA triggers mitochondrial pathway through inhibition of survival pathways, for example (**1**) downregulation of ERK1/2 by PPARγ, and (**2**) Akt/GSK-3β inhibition by PTEN activation. Moreover, (**3**) LA causes oxidative stress by massive ROS accumulation and GSH decrease. Likewise, (**4**) LA generates accumulation of intracellular Ca^2+^, and expression of the UPR-associated genes (CHOP, GRP78, and GRP94), (**5**) downregulation of the PGE2 production and telomerase activity through suppressing COX-2 and hTERT expression. (**6**) All these events cause loss of mitochondrial membrane potential, increase in the Bax and Bad expression, and downregulate expression of Bcl-2, producing MOMP and promote the release of cytochrome c from the mitochondria, causing activation of caspase-9 and caspase-3, and decreasing ATP level. (**7**) Additionally, other studies demonstrated the upregulation of p21 and p53 mRNA. (**B**) (**8**) Regarding extrinsic pathway LA increases FAS and caspase-8 levels. LA: Linoleic Acid. ERK1/2: Extracellular signal-regulated protein kinases 1 and 2. PPARγ: Peroxisome proliferator-activated receptor-gamma. Akt: Protein kinase B. GSK-3β: Glycogen Synthase Kinase 3 Beta. PTEN: Phosphatase and tensin homolog. ROS: Reactive oxygen species. GSH: Glutathione. UPR: Unfolded protein response. CHOP: C/EBP Homologous Protein. GRP78: Glucose-Regulated Protein 78. GRP94: Glucose-Regulated Protein 94. PGE2: Prostaglandin E2. COX-2: Cyclooxygenase 2. hTERT: Telomerase reverse transcriptase. Bax: Bcl-2 Associated X-protein. Bad: Bcl-2 associated agonist of cell death. Bcl-2: B-cell lymphoma 2. MOMP: Mitochondrial outer membrane permeabilization. ATP: Adenosine triphosphate. p21: Cyclin-dependent kinase inhibitor 1A. p53: Tumor protein P53. FAS: Fas cell surface death receptor.

**Figure 6 ijms-24-11691-f006:**
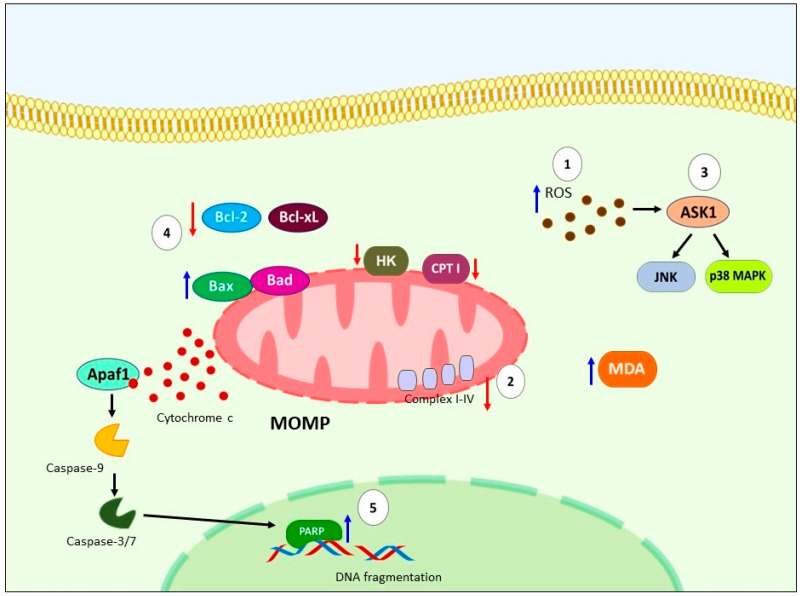
Schematic representation of molecular apoptosis pathways regulates by GLA. (**1**) GLA causing oxidative stress by ROS accumulation and lipid peroxidation, additionally, (**2**) has been report downregulation of mitochondrial CPT I, hexoquinase, and mitochondrial respiratory chain complexes I+III and IV, high levels of MDA. (**3**) Another mechanism reported is ROS/ASK1/JNK/p38 MAPK pathway. (**4**) All these events trigger intrinsic apoptosis by downregulation of Bcl-Xl, Bcl-2, upregulation of Bad, causing loss of mitochondrial membrane potential, and subsequent release of cytochrome c, activation of caspase-9 and -3, (**5**) leading to cleavage of PARP and DNA fragmentation. GLA: Gamma-linolenic acid. ROS: Reactive oxygen species. CPT I: carnitine palmitoyltransferase I. MDA: malondialdehyde. ASK1: Apoptosis signal-regulating kinase 1. JNK: c-Jun N-terminal kinase. P38 MAPK: p38 MAP Kinase. Bcl-Xl: B-cell lymphoma-extra-large. Bcl-2: Bad: Bcl-2 associated agonist of cell death. PARP: Poly (ADP-ribose) polymerase. DNA: Deoxyribonucleic acid.

**Figure 7 ijms-24-11691-f007:**
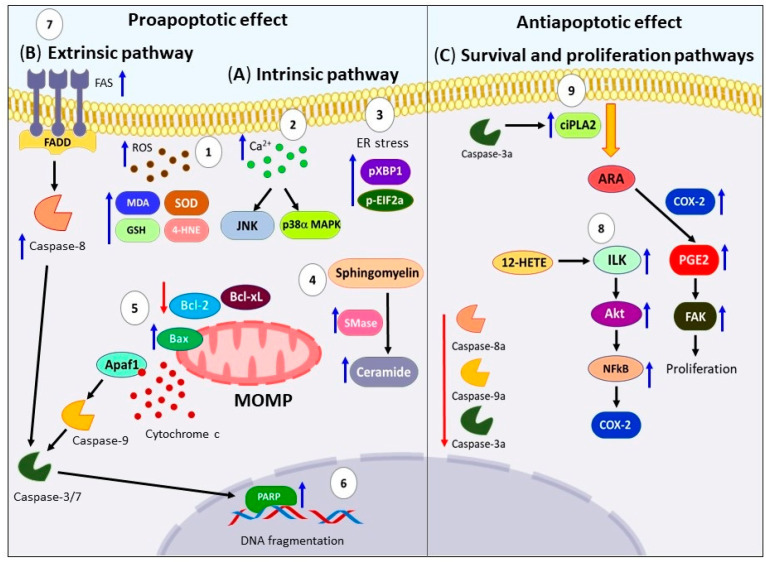
Effects of ARA on apoptotic death pathways. (**A**) (**1**) Treatment with ARA induces an increase in oxidative stress through the production of ROS and lipid peroxidation. This leads to elevated levels of MDA, 4-HNE, SOD, and GSH-PX activity. (**2**) Furthermore, the elevated ROS levels and intracellular calcium concentration activate the p38α MAPK and JNK1 pathways. (**3**) ARA also triggers ER stress, characterized by high levels of pXBP1 and phosphorylated eIF2α (p-eIF2α). (**4**) Another mechanism involves the accumulation of unmetabolized ARA, which activates the enzyme SMase, resulting in elevated ceramide levels—an important second messenger and potent activator of apoptosis. (**5**) These mechanisms collectively lead to the association of Bax protein with mitochondria, downregulation of Bcl-xL and Bcl-2, loss of mitochondrial membrane potential, release of cytochrome c from the mitochondria to the cytosol, and activation of caspase-9 and -3. (**6**) This cascade ultimately results in the breakdown of PPAR and DNA fragmentation. (**B**) (**7**) In the extrinsic pathway, ARA increases the expression of FAS, caspases-8, and -3. (**C**) On the other hand, ARA induces anti-apoptotic effects by activating survival and proliferation pathways, such as the Akt axis. (**8**) Additionally, 12-HETE triggers the activation of the ILK/NF-κB axis. (**9**) Another novel mechanism suggests that during apoptosis, active caspase-3 can activate ciPLA2, leading to the release of ARA and the production of PGE2. This abnormal activation of FAK promotes proliferation. Furthermore, downregulation of active caspases 3, 8, and 9 has been reported. ARA: Arachidonic acid. ROS: Reactive oxygen species. MDA: Malondialdehyde. 4-HNE: 4-hydroxy-2-nonenal. SOD: superoxide dismutase. GSH-PX: Glutathione peroxidase. p38α MAPK: p38α MAP Kinase. JNK1: c-Jun N-terminal kinase 1. ER: endoplasmic reticulum. pXBP1: processed form of XBP1. p-eIF2α: phosphorylated eukaryotic initiation factor-2α. SMase: sphingomyelinase. Bcl-xL: B-cell lymphoma-extra-large. Bcl-2: B-cell lymphoma 2. PPAR: Poly (ADP-ribose) polymerase. DNA: deoxyribonucleic acid. FAS: Fas cell surface death receptor. Akt: Protein kinase B. 12-HETE: 12-Hydroxyeicosatetraenoic acid. ILK: integrin-linked kinase. NF-κB: nuclear factor κB. ciPLA2: calcium-independent phospholipase A2. PGE2: Prostaglandin E2. FAK: Focal adhesion kinase.

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
