# Peer review of "The Involvement of Polyunsaturated Fatty Acids in Apoptosis Mechanisms and Their Implications in Cancer"

_ijms, 2023, doi:10.3390/ijms241411691_

Round 1

Reviewer 1 Report

The authors reflect on the role of polyunsaturated fatty acids in apoptotic pathways in their review. In order to elucidate their involvement in cancer, they detailed the role of each PUFA in the molecular pathway regulating apoptosis in cancer. The review highlights the importance of understanding the mechanisms behind apoptosis in cancer cells, as it can lead to the development of new therapeutic strategies. Moreover, the authors discuss the potential of PUFA supplementation as an adjuvant therapy in cancer treatment. 

The work is well explained, but some improvements are needed. 

  • 1) It is advised to conduct a literature review because many of the articles cited are out-of-date and many recent papers haven't been cited. 

  • 2) The authors reported that western population have a high consumption rate of omega-6 to omega-3 PUFAs. How the western population could be crucial to study the effects of these PUFAs on apoptosis pathways? It is not sufficiently clear and argued the role of omega-6 PUFAs in this population, in the references 14 and 123 the consumption of PUFAs in this population is correlated with obesity, cancers and inflammation meanwhile the authors reported a pro-apoptotic effect and consequently an anti-cancer effect of PUFAs. How the authors explain this apparent contradiction? 

  • 3) It is strongly recommended that authors provide a list of the abbreviations used in the figure legend. 

  • 4) It is recommended that authors strengthen their arguments in the introductory portion of the second paragraph in order to contextualize that section with the paragraphs that follow. 

  • 5) It is not clear from Figures 2 through 7 at which level of the apoptotic pathway the various PUFAs act or how they do so; the authors should provide this information in the appropriate figures. 

  • 6) It is recommended that additional information be included, as well as that the role of ALA in apoptosis be expounded upon. 

  • 7) The authors' statement at lines 388-390 appears to be conflicting. Is LA implicated in cancer initiation, or does it promote apoptosis? Please clarify or amend this statement. Please include a citation. 

  • 8) It is recommended that the authors standardize the text. When you read it, it is clear that different persons contributed to the writing of the various paragraphs, just as the settings of the various paragraphs are distinct from one another. 

  • 9) According to the authors, in what ways do PUFAs have the potential to constitute a novel and promising strategy for the treatment of cancer?

Author Response

Answer to the Reviewer Comments Manuscript ID ijms-2465707

Reviewer 1:

Comments and Suggestions for Authors

The authors reflect on the role of polyunsaturated fatty acids in apoptotic pathways in their review. In order to elucidate their involvement in cancer, they detailed the role of each PUFA in the molecular pathway regulating apoptosis in cancer. The review highlights the importance of understanding the mechanisms behind apoptosis in cancer cells, as it can lead to the development of new therapeutic strategies. Moreover, the authors discuss the potential of PUFA supplementation as an adjuvant therapy in cancer treatment. 

The work is well explained, but some improvements are needed. 

Question 1: It is advised to conduct a literature review because many of the articles cited are out-of-date and many recent papers haven't been cited. 

Author response a: Thank you very much for the suggestion. We added the follow recently articles about the subject throughout the text:

  • LA induces ROS production on HepG2 Cells (Line 466). F. S. Teixeira, L. L. Pimentel, S. S. M. P. Vidigal, J. Azevedo-Silva, M. E. Pintado, and L. M. Rodríguez-Alcalá, “Differential Lipid Accumulation on HepG2 Cells Triggered by Palmitic and Linoleic Fatty Acids Exposure,” Molecules, vol. 28, no. 5, 2023.
  • DHA has been shown to stimulate ROS generation in various cell lines, including docetaxel-resistant PC3 prostate cancer cells (Line 317). Z. C. Shao et al., “Docosahexaenoic Acid Reverses Epithelial-Mesenchymal Transition and Drug Resistance by Impairing the PI3K/AKT/ Nrf2/GPX4 Signalling Pathway in Docetaxel-Resistant PC3 Prostate Cancer Cells,” Folia Biol. (Praha)., vol. 68, no. 2, pp. 59–71, 2022.
  • DHA can induce apoptosis on a gastric adenocarcinoma-derived cell line (AGS). (Line 350). L. Ortega et al., “The Ω-3 fatty acid docosahexaenoic acid selectively induces apoptosis in tumor-derived cells and suppress tumor growth in gastric cancer,” Eur. J. Pharmacol., vol. 896, no. August 2020, 2021.
  • Several authors have suggested the use of omega-3 like adjuvant therapy in the treatment of cancer patients. Among the effects of omega-3 supplementation in the diet of cancer patients with chemotherapy are treatment tolerance, decrease in the tumor and lean body mass gain (Line 704-706). de Freitas Rodrigues, H. K. Philippsen, M. F. Dolabela, C. Y. Nagamachi, and J. C. Pieczarka, “The Potential of DHA as Cancer Therapy Strategies: A Narrative Review of In Vitro Cytotoxicity Trials,” Nutrients, vol. 15, no. 8, pp. 1–21, 2023.

Question 2: The authors reported that western population have a high consumption rate of omega-6 to omega-3 PUFAs. How the western population could be crucial to study the effects of these PUFAs on apoptosis pathways? It is not sufficiently clear and argued the role of omega-6 PUFAs in this population, in the references 14 and 123 the consumption of PUFAs in this population is correlated with obesity, cancers and inflammation meanwhile the authors reported a pro-apoptotic effect and consequently an anti-cancer effect of PUFAs. How the authors explain this apparent contradiction? 

Author response a:  We have clarified this contradiction in new version of the manuscript (Lines 53-59), we should mention that the modern western diet (20:1) contains an excess of omega-6 PUFAs compared to Mediterranean diets (4:1), that is due to agribusiness and modern agriculture and have occurred in parallel with a significant increase in the prevalence of overweight, obesity and cancer.

  • A. Albar, “Dietary Omega-6/Omega-3 Polyunsaturated Fatty Acid (PUFA) and Omega-3 Are Associated with General and Abdominal Obesity in Adults: UK National Diet and Nutritional Survey,” Cureus, vol. 14, no. 10, 2022.
  • Simopoulos, A.P. Evolutionary Aspects of Diet: The Omega-6/Omega-3 Ratio and the Brain. Mol. Neurobiol. 2011, 44, 203–215.

Question 3: It is strongly recommended that authors provide a list of the abbreviations used in the figure legend. 

Author response a:  We have included the abbreviations in all figure legends in the new version of the manuscript.

Question 4: It is recommended that authors strengthen their arguments in the introductory portion of the second paragraph in order to contextualize that section with the paragraphs that follow. 

Author response a: We have already connected better the two paragraphs

Question 5:  It is not clear from Figures 2 through 7 at which level of the apoptotic pathway the various PUFAs act or how they do so; the authors should provide this information in the appropriate figures. 

Author response a: We have included a serial number in each figure, to explain which pathways are involved in each case.

Question 6:  It is recommended that additional information be included, as well as that the role of ALA in apoptosis be expounded upon. 

Author response a: The information the roll of ALA in apoptosis mechanisms is already included in the manuscript (Lines 121-172 and described in Figure 2).

Question 7: The authors' statement at lines 388-390 appears to be conflicting. Is LA implicated in cancer initiation, or does it promote apoptosis? Please clarify or amend this statement. Please include a citation. 

Author response a: The reviewer is right; we modified the text about the involvement of LA in the mechanisms of apoptosis and tumor initiation (Lines 451-453). In order to clarify this important point.

Question 8: It is recommended that the authors standardize the text. When you read it, it is clear that different persons contributed to the writing of the various paragraphs, just as the settings of the various paragraphs are distinct from one another. 

Author response a: We have homogenized the follow of each paragraphs.

Question 9: According to the authors, in what ways do PUFAs have the potential to constitute a novel and promising strategy for the treatment of cancer?

Author response a: As part of our conclusions (Lines 697-719), We also emphasize the importance of maintaining a balance between omega-6 and omega-3 fatty acid consumption, as it can lead to favorable biological outcomes, including decreased inflammation and the induction of apoptosis, which have a beneficial effect in preventing cancer development. Furthermore, the use of omega-3 poly-unsaturated fatty acids (PUFAs) as adjuvant therapy in cancer treatment has been suggested by various researchers. Omega-3 supplementation in the diets of cancer patients undergoing chemotherapy has shown potential benefits, such as improved treatment tolerance, decreased tumor size, and increased lean body mass [181]. For example, in a randomized trial with 11 colorectal cancer patients, the group that ingested 2 g/day of fish oil exhibited significant increases in EPA and DHA levels in blood plasma (1.8 and 1.4 times higher, respectively), along with muscle mass gain (mean of +1.2 kg). This group also demonstrated greater chemotherapy tolerance and a reduction in tumor size compared to the untreated group [182]. In another study, 128 patients with gastrointestinal cancer and cachexia were provided with a diet containing or lacking 1.1 g of EPA, 0.5 g of DHA, and 16 g of protein. The patients were monitored using bioelectrical impedance analysis, and the authors observed that the fish oil-enriched diet contributed to improved chemotherapy tolerance, tumor shrinkage, and increased lean body mass over time [183].

Importantly, the use of enzyme inhibitors to enhance the accumulation of specific oxylipins, exerting pro-apoptotic effects, has shown promise in in vitro and murine models [184], [185]. These findings indicate the potential of such inhibitors as powerful tools in cancer treatment. However, further research and clinical trials are necessary to validate their efficacy and safety in human patients.

  • de Freitas Rodrigues, H. K. Philippsen, M. F. Dolabela, C. Y. Nagamachi, and J. C. Pieczarka, “The Potential of DHA as Cancer Therapy Strategies: A Narrative Review of In Vitro Cytotoxicity Trials,” Nutrients, vol. 15, no. 8, pp. 1–21, 2023.
  • C. Mocellin et al., “Fish oil decreases C-reactive protein/albumin ratio improving nutritional prognosis and plasma fatty acid profile in colorectal cancer patients,” Lipids, vol. 48, no. 9, pp. 879–888, Sep. 2013.
  • Shirai et al., “Fish oil-enriched nutrition combined with systemic chemotherapy for gastrointestinal cancer patients with cancer cachexia,” Sci. Rep., vol. 7, no. 1, Dec. 2017.
  • Guodong et al., “Epoxy metabolites of docosahexaenoic acid ( DHA ) inhibit angiogenesis , tumor growth , and metastasis,” Pnas, vol. 110, no. 16, pp. 6530–6535, 2013.
  • Zhang et al., “Dual inhibition of cyclooxygenase-2 and soluble epoxide hydrolase synergistically suppresses primary tumor growth and metastasis,” Proc. Natl. Acad. Sci. U. S. A., vol. 111, no. 30, pp. 11127–11132, 2014

Reviewer 2 Report

This paper is a review paper on the apoptosis efficacy of PUFAs in cancer, which has been fully explained and well written in an appropriate picture. Many people are interested in the apoptosis of PUFAs and are thought to be efficiently used for prevention and treatment in the future.

I think a better paper will be completed if you add answers to some questions.

1.This paper describes apoptosis well. However, it would be nice to mention Autophagy, which is highly related to apoptosis, even briefly.

2.There are many reports that ion channels are also involved in the regulation of apoptosis. Is there no study that PUFAs controls apoptosis by affecting ion channels? If there is, you would like to mention it briefly.

3. It would be nice to add advice related to daily clinical studies, such as how we can get PUFAs in real life.

4. Are PUFAs involved in all cancer cell death in our bodies? Or does it work better for certain organ cancers?

5. Between the proapoptotic effect and the antiapoptotic effect of PUFAs, which result does the author think is more reasonable and has more evidence?

Author Response

Answer to the Reviewer Comments Manuscript ID ijms-2465707

Reviewer 2:

Comments and Suggestions for Authors

This paper is a review paper on the apoptosis efficacy of PUFAs in cancer, which has been fully explained and well written in an appropriate picture. Many people are interested in the apoptosis of PUFAs and are thought to be efficiently used for prevention and treatment in the future.

I think a better paper will be completed if you add answers to some questions.

Question 1: This paper describes apoptosis well. However, it would be nice to mention Autophagy, which is highly related to apoptosis, even briefly.

Author response a: Thank you for your feedback. In our current study, we wanted to specifically focus on the apoptotic process. However, we appreciate your suggestion and will consider conducting a separate review on the role of polyunsaturated fatty acids (PUFAs) in autophagy in the future. This topic warrants further exploration and understanding, and we believe it would be valuable to investigate the effects of PUFAs on autophagy in the context of cancer and other diseases.

Question 2: There are many reports that ion channels are also involved in the regulation of apoptosis. Is there no study that PUFAs controls apoptosis by affecting ion channels? If there is, you would like to mention it briefly.

Author response a:  Thank you for important suggestion. It provides valuable information on the role of polyunsaturated fatty acids (PUFAs) in regulating ion channels and their potential implications in cancer development. Here is the paragraph that we have included in the new version of the manuscript:

Other studies have also suggested that the participation of PUFAs as potent modulators of animal ion channels. They can form micelles that can fuse with the lipid bilayer, influencing cell membrane organization and altering channel function. PUFAs are known to increase membrane fluidity and screen surface charges, resulting in a shift in the voltage-dependence of the channels. Particularly, ALA can cause accumulation and increase in intracellular calcium ion (Ca2+) levels, generating arrest in the sub-G1 phase and triggers the pro-apoptotic process. Additionally, it has been observed that ALA provides a downregulation of voltage-dependent anion channel (VDAC) present in the outer mitochondrial membrane, support the mitochondrial apoptosis (Lines 132-140)

This revised paragraph provides a comprehensive explanation of how PUFAs can modulate ion channels, highlighting their potential role in cancer and emphasizing the significance of specific PUFA characteristics in their interaction with ion channels.

  • Fiske, J.L.; Fomin, V.P.; Brown, M.L.; Duncan, R.L.; Sikes, R.A. Voltage-Sensitive Ion Channels and Cancer. Cancer Metastasis Rev. 2006, 25, 493–500
  • Y. Kim, H. D. Park, E. Park, J. W. Chon, and Y. K. Park, “Growth-inhibitory and proapoptotic effects of alpha-linolenic acid on estrogen-positive breast cancer cells: Second look at n-3 fatty acid,” in Annals of the New York Academy of Sciences, 2009, vol. 1171, pp. 190–195.
  • C. Zhang, G. Gdynia, V. Ehemann, and W. Roth, “The HMGB1 protein sensitizes colon carcinoma cells to cell death triggered by pro-apoptotic agents,” Int. J. Oncol., vol. 46, no. 2, pp. 667–676, Feb. 2015.
  • Roy et al., “Alpha-linolenic acid stabilizes HIF-1 α and downregulates FASN to promote mitochondrial apoptosis for mammary gland chemoprevention,” Oncotarget, vol. 8, no. 41, pp. 70049–70071, 2017.

Question 3: It would be nice to add advice related to daily clinical studies, such as how we can get PUFAs in real life.

Author response a: We have included the following information in the section of Concluding Remarks and Future Perspectives: Several authors have suggested the use of omega-3 polyunsaturated fatty acids (PUFAs) as adjuvant therapy in the treatment of cancer patients. Omega-3 supplementation in the diet of cancer patients undergoing chemotherapy has shown various beneficial effects, including improved treatment tolerance, decreased tumor size, and increased lean body mass. For instance, in a randomized trial involving 11 colorectal cancer patients, the group receiving 2 g/day of fish oil exhibited significant increases in EPA and DHA levels in blood plasma (1.8 and 1.4 times higher, respectively), along with muscle mass gain (+1.2 kg). This group also demonstrated greater chemotherapy tolerance and a reduction in tumor size compared to the untreated group. Another study involving 128 patients with gastrointestinal cancer and cachexia provided a diet enriched with 1.1 g of EPA, 0.5 g of DHA, and 16 g of protein. The patients were monitored using bioelectrical impedance analysis, and the authors observed that the fish oil-enriched diet contributed to improved chemotherapy tolerance, tumor shrinkage, and increased lean body mass over time (Lines 690-713).

This added information highlights the positive effects of omega-3 PUFA supplementation in cancer patients undergoing chemotherapy, emphasizing the benefits in terms of treatment tolerance, tumor reduction, and lean body mass gain.

  • de Freitas Rodrigues, H. K. Philippsen, M. F. Dolabela, C. Y. Nagamachi, and J. C. Pieczarka, “The Potential of DHA as Cancer Therapy Strategies: A Narrative Review of In Vitro Cytotoxicity Trials,” Nutrients, vol. 15, no. 8, pp. 1–21, 2023.
  • Mocellin, M.C.; Silva, J.D.A.P.E.; Camargo, C.; Fabre, M.E.D.S.; Gevaerd, S.; Naliwaiko, K.; Moreno, Y.M.F.; Nunes, E.; Trindade, E.B.S.D.M. Fish Oil Decreases C-Reactive Protein/Albumin Ratio Improving Nutritional Prognosis and Plasma Fatty Acid Profile in Colorectal Cancer Patients. Lipids 2013, 48, 879–888.
  • Shirai, Y.; Okugawa, Y.; Hishida, A.; Ogawa, A.; Okamoto, K.; Shintani, M.; Morimoto, Y.; Nishikawa, R.; Yokoe, T.; Tanaka, K.; et al. Fish oil-enriched nutrition combined with systemic chemotherapy for gastrointestinal cancer patients with cancer cachexia. Sci. Rep. 2017, 7, 4826.

Question 4: Are PUFAs involved in all cancer cell death in our bodies? Or does it work better for certain organ cancers?

Author response a:  Thank you for your question. At present, there is limited available data to provide a definitive answer to this inquiry. However, a particular study includes a figure illustrating the preferred sites for the accumulation and expression of polyunsaturated fatty acids (PUFAs). Additionally, there is evidence indicating that PUFAs are vital components of cell membranes, strongly suggesting their presence throughout the body, including the brain. This implies that they may play a role in cancer cell death across various organs. Based on the aforementioned information, it is reasonable to speculate that the mechanisms by which PUFAs contribute to apoptosis and cell death may have a preferential role in organs where they are more abundant. However, further research is required to gain a comprehensive understanding of the specific roles and mechanisms of PUFAs in cancer cell death in different body organs.

  • Zhang H, He Y, Song C, Chai Z, Liu C, Sun S, Huang Q, He C, Zhang X, Zhou Y, Zhao F. Analysis of fatty acid composition and sensitivity to dietary n-3 PUFA intervention of mouse n-3 PUFA-enriched tissues/organs. .Prostaglandins Leukot Essent Fatty Acids. 2023 May;192:102568.
  • Patterson, E.; Wall, R.; Fitzgerald, G.F.; Ross, R.; Stanton, C. Health Implications of High Dietary Omega-6 Polyunsaturated Fatty Acids. J. Nutr. Metab. 2012, 2012, 539426.

Question 5: Between the proapoptotic effect and the antiapoptotic effect of PUFAs, which result does the author think is more reasonable and has more evidence?

Author response a: Based on the comprehensive information presented in this review, we have a strong belief that omega-3 polyunsaturated fatty acids (PUFAs) exert a clear pro-apoptotic effect. On the other hand, omega-6 PUFAs, particularly arachidonic acid (ARA), exhibit a dual effect, including both pro-apoptotic and pro-survival outcomes. However, these behaviors appear to vary depending on the specific in vitro and in vivo models studied, the concentration of fatty acids used, the duration of incubation, and the combination of treatments with other drugs.

These findings highlight the complexity of the effects of PUFAs on apoptosis, suggesting that the specific outcomes are influenced by various factors. The context-dependent nature of these effects underscores the need for further research to elucidate the precise mechanisms underlying the diverse actions of PUFAs in apoptosis regulation. By considering these factors, future studies can provide valuable insights into optimizing the therapeutic potential of PUFAs in cancer treatment.

This information has been incorporated into the section of Concluding Remarks and Future Perspectives (Lines 682-695) to provide a comprehensive overview of the varied effects of omega-3 and omega-6 PUFAs on apoptosis and suggest avenues for further investigation.

Reviewer 3 Report

 The manuscript provides a well written overview on the role of PUFA in apoptosis. There are several recent research papers that are not cited in the review. This reviewer is aware of the fact that it is not possible to cite all of them because of their large number. But maybe the authors could comment on this and briefly describe how they selected papers (e.g. date of publications; query terms for a pubmed or other data base search).

Minor comments:

line 76: use small letter "c" in "cytochrome c"

Figure 6: it would be better to place the respiratory chain complexes at the inner mitochondrial membrane (cristae)

Figure 7: "Sphingomielin" should be "Sphingomyelin"

There are several typos/grammar errors/incomplete sentences, for example:

line 129/130: "....induces a increased levels of ..."

line 161: "Moreover, have been described ..."

line 199/200: "..., none of the death ..." (meaning unclear)

Author Response

Answer to the Reviewer Comments Manuscript ID ijms-2465707

Reviewer 3:

Comments and Suggestions for Authors

The manuscript provides a well written overview on the role of PUFA in apoptosis. There are several recent research papers that are not cited in the review. This reviewer is aware of the fact that it is not possible to cite all of them because of their large number. But maybe the authors could comment on this and briefly describe how they selected papers (e.g. date of publications; query terms for a pubmed or other data base search).

Minor comments:

Question 1: line 76: use small letter "c" in "cytochrome c"

Author response a: Thank you for your comment. We have corrected this error.

Question 2: Figure 6: it would be better to place the respiratory chain complexes at the inner mitochondrial membrane (cristae).

Author response a: We have modified the respiratory chain complexes located in the inner mitochondrial membrane.

Regenerate response

Question 3: Figure 7: "Sphingomielin" should be "Sphingomyelin"

Author response a:  We have corrected this error in the Figure 7.

Comments on the Quality of English Language

There are several typos/grammar errors/incomplete sentences, for example:

Question 4: line 129/130: "....induces a increased levels of ..."

Author response a: We have changed the word from “induces a increased” to “induces an increased”. In addition, we go over the entire manuscript to identify and correct more typos/grammar error.

Question 5: line 161: "Moreover, have been described ..."

Author response a:  We have changed the phrase from "Moreover, have been described ..." to “Moreover, studies have shown”.

Question 6: line 199/200: "..., none of the death ..." (meaning unclear)

Author response a:  We have changed the phrase in order to be clearer.

Round 2

Reviewer 1 Report

I would like to thank the authors for answering all my requests.

Reviewer 2 Report

It is well revised